# Tomato-Thaumatin-like Protein Genes *Solyc08g080660* and *Solyc08g080670* Confer Resistance to Five Soil-Borne Diseases by Enhancing β-1,3-Glucanase Activity

**DOI:** 10.3390/genes14081622

**Published:** 2023-08-14

**Authors:** Xinyun Li, Bin Xu, Junqiang Xu, Zuosen Li, Caiqian Jiang, Ying Zhou, Zhengan Yang, Minghua Deng, Junheng Lv, Kai Zhao

**Affiliations:** College of Horticulture and Landscape, Yunnan Agricultural University, Kunming 650201, China; 15126719712@163.com (X.L.); yangzhengan@ynau.edu.cn (Z.Y.); dengminghua2013@sina.com (M.D.); junhenglv@163.com (J.L.)

**Keywords:** bacterial pathogens, disease resistance, fungal pathogens, soil-borne disease, thaumatin-like protein

## Abstract

Although thaumatin-like proteins (*TLPs*) are involved in resistance to a variety of fungal diseases, whether the *TLP*5 and *TLP*6 genes in tomato plants (*Solanum lycopersicum*) confer resistance to the pathogenesis of soil-borne diseases has not been demonstrated. In this study, five soil-borne diseases (fungal pathogens: *Fusarium solani*, *Fusarium oxysporum*, and *Verticillium dahliae*; bacterial pathogens: *Clavibacter michiganense* subsp. *michiganense* and *Ralstonia solanacearum*) were used to infect susceptible “No. 5” and disease-resistant “S-55” tomato cultivars. We found that *SlTLP5* and *SlTLP6* transcript levels were higher in susceptible cultivars treated with the three fungal pathogens than in those treated with the two bacterial pathogens and that transcript levels varied depending on the pathogen. Moreover, the *SlTLP5* and *SlTLP6* transcript levels were much higher in disease-resistant cultivars than in disease-susceptible cultivars, and the *SlTLP5* and *SlTLP6* transcript levels were higher in cultivars treated with the same fungal pathogen than in those treated with bacterial pathogens. *SlTLP6* transcript levels were higher than *SlTLP5*. *SlTLP5* and *SlTLP6* overexpression and gene-edited transgenic mutants were generated in both susceptible and resistant cultivars. Overexpression and knockout increased and decreased resistance to the five diseases, respectively. Transgenic plants overexpressing *SlTLP5* and *SlTLP6* inhibited the activities of peroxidase (POD), superoxide dismutase (SOD), ascorbate peroxidase (APX), and catalase (CAT) after inoculation with fungal pathogens, and the activities of POD, SOD, and APX were similar to those of fungi after infection with bacterial pathogens. The activities of CAT were increased, and the activity of β-1,3-glucanase was increased in both the fungal and bacterial treatments. Overexpressed plants were more resistant than the control plants. After *SlTLP5* and *SlTLP6* knockout plants were inoculated, POD, SOD, and APX had no significant changes, but CAT activity increased and decreased significantly after the fungal and bacterial treatments, contrary to overexpression. The activity of β-1,3-glucanase decreased in the treatment of the five pathogens, and the knocked-out plants were more susceptible to disease than the control. In summary, this study contributes to the further understanding of *TLP* disease resistance mechanisms in tomato plants.

## 1. Introduction

Plants have evolved a range of components to fight pathogens, including the expression of pathogenesis-related (PR) proteins in response to complex and diverse environments. Currently, at least 17 PR proteins are known to be induced by oomycetes, fungi, bacteria, nematodes, viruses, and viroids, as well as insect bites [1,2]. As a PR plant disease-resistance protein, the PR protein has been used widely in crop protection against fungi [3]. The anti-fungal activity of thaumatin-like protein (TLP) has been studied intensively. Plant *TLP* mainly inhibits pathogenic and non-pathogenic fungi by lysing fungal spores, inhibiting spore germination, and reducing the vitality of young mycelium [4,5]. Recently, *TLP* genes were found to confer resistance to pathogenic fungi, including sclerotia, powdery mildew, and *Pseudomonas syringae* [6,7]. *TLP* transgenic plants can delay the development of a variety of fungal diseases and enhance plant resistance to pathogenic fungi [8,9]. Furthermore, *TLP* genes can be induced by a variety of biotic and abiotic stresses [10,11,12,13,14,15].

Fungicides are applied to control plant diseases in traditional agriculture. However, the widespread use of pesticides may result in serious environmental pollution and food safety problems. *TLP* genes have been successfully expressed in a variety of plants, and indica rice cultivars overexpressing *TLP*s enhance resistance to *Rhizoctonia solani*, a pathogen of rice sheath blight disease, and *Sarocladium oryzae*, a pathogen of rice sheath rot [16]. Activity studies of recombinant *Solanum nigrum TLPs* in vitro have demonstrated that they are equally sensitive to *Fusarium solani* f. sp. *glycines*, *Colletotrichum* spp., *Macrophomina phaseolina*, and *Phytophthora nicotianae* var. *parasitica* [17] and exhibited resistance to a variety of pathogenic fungi in transgenic plants. Given the broad resistance of *TLPs*, breeding *TLP* transgenic plants is an effective approach to achieving plant disease resistance.

A disease-resistance gene cluster was identified from chromosome 8 of the inbred line CLN2037E by our project team. The cluster contained six genes, five of which were *TLP* genes, including *Solyc08g080660* and *Solyc08g080670*, referred to as *SlTLP5* and *SlTLP6*, respectively. The overexpression and knockout of *SlTLP5* and *SlTLP6* corresponded to an increase and decrease in resistance to late blight, respectively [18]. The role of *SlTLP5* and *SlTLP6* in the defense signal transduction of tomato plants after infection with five soil-borne diseases was illustrated in our study. Five soil-borne diseases were used to infect resistant “S-55” and susceptible “No. 5” tomato cultivars, and the transcription levels of *SlTLP5* and *SlTLP6* in resistant and susceptible cultivars were determined via quantitative real-time PCR (RT-qPCR). Transgenic plants overexpressing *SlTLP5* and *SlTLP6* and gene-edited transgenic plants were generated in susceptible and resistant cultivars, respectively, to identify the disease resistance of transgenic plants.

## 2. Materials and Methods

### 2.1. Experimental Material and Bacterial Infection

The susceptible “No. 5” and resistant “S-55” inbred tomato cultivars were developed and preserved by our research group, and the plants were cultivated in growth chambers with light for 16 h (28 °C) and without light for 8 h (20 °C). The strains *Fusarium solani*, *Fusarium oxysporum*, *Verticillium dahliae*, *Clavibacter michiganense* subsp. *michiganense*, and *Ralstonia solanacearum* were obtained from the Institute of Vegetables and Flowers, Chinese Academy of Agricultural Sciences (Kunming, Yunnan Province, China). The strains *R. solanacearum* and *C. michiganense* subsp. *michiganense* were cultured in nutrient broth at 28 °C with a rotational speed of 150 rpm. After 48 h, they were washed three times via centrifugation with sterile water, and *R. solanacearum* and *F. oxysporum* suspensions were resuspended in deionized water and adjusted to OD_600_ = 0.1 (10^8^ cfu·mL^−1^) using a spectrophotometer (Shimadzu UV-2600, Kyoto, Japan). The pathogens *F. oxysporum*, *V. dahliae*, and *F. solani* were cultured in PDA (Oxoid, Basingstoke, UK) medium for 4–6 days at 28 °C under an oscillation of 200 rpm. The mycelium was filtered off with four layers of gauze; the suspension was diluted with sterile distilled water to a suitable concentration (calculated using a hemocytometer plate-counting method). The inoculated *F. oxysporum* and *V. dahliae* bacterial solutions had a concentration of 1 × 10^9^ cfu·mL^−1^, and the inoculated *F. solani* bacterial solution had a concentration of 1 × 10^8^ cfu·mL^−1^. Then, 50 mL of each of the five pathogen suspensions was poured onto seven leaves of tomato seedlings in each pot, and those at d 0 were used as the control group. The inoculated seedlings were placed in a plant growth chamber and preserved at 100% relative humidity (RH) and 20 ± 1 °C dark conditions for 24 h. Subsequently, RH was lowered to 60% from 80% under light for 14 h/d [19]. The tomato leaves were frozen in liquid nitrogen after treatment for 0 (control), 3, 5, 7, and 9 days, and then stored at −80°C.

### 2.2. Identification of Disease Resistance in Tomato Leaves via RT-qPCR

Total RNA was extracted from tomato leaves of the disease-susceptible and disease-resistant inbred lines “No. 5” and “S-55” after isolation and treatment using the Huayuoyang Rapid Universal Plant RNA Extraction Kit (Huayuoyang Biotechnology Co., Ltd., Beijing, China), and the quality and quantity of extracted RNA were confirmed using 1.5% (*w*/*v*) agar gel electrophoresis and NanoDrop 1000 spectrophotometry (Thermo Fisher Scientific, New York, NY, USA). Two micrograms of total RNA were used to synthesize cDNA using TransScript One-Step gDNA Removal and cDNA Synthesis SuperMix (TransGen, Beijing, China) according to the manufacturer’s protocol. The primers were designed using the Primer Premier 6 software (Premier, San Francisco, CA, USA) and synthesized by Tsingke Biotechnology (Qingke Biotechnology Co., Ltd., Beijing, China). RT-qPCR was performed with a CFX96 PCR machine (Bio-Rad, Hercules, CA, USA) using a SYBR Green PCR Master Mix (TransGen). The 2^−ΔΔCT^ method [20] was used to calculate relative mRNA abundance. The tomato housekeeping gene ribosomal protein L2 [20,21] was used as an internal control, and each value represented the mean of three biological replicates.

### 2.3. Generation of Overexpressing Transgenic Tomato Plants

The function of *SlTLP5* and *SlTLP6* was verified using an overexpression technique. The “No. 5” tomato inbred line was used for overexpression experiments.

Amplification primers for the CDS region of *SlTLP5* and *SlTLP6* were designed using the Primer6 software, and BamHI and SacI restriction sites were introduced to each primer, respectively. The *SlTLP5* and *SlTLP6* genes were ligated into the binary vector pBI121. Leaf disc transformation was used for transformation [22], and explants were obtained from the cotyledons of one-week-old seedlings. The kanamycin and RT-PCR methods were used to screen transgenic plant strains. The plants overexpressing *SlTLP5* and *SlTLP6* were validated using two pairs of specific primers and universal primers. The expression levels of *SlTLP5* and *SlTLP6* in transgenic lines were detected using RT-qPCR.

### 2.4. Generation of Gene-Edited Tomato Plants

The Clustered Regularly Interspaced Short Palindromic Repeats (CRISPR)/Cas9 system was used to generate *SlTLP5* and *SlTLP6* knockout strains in the resistant tomato cultivar “S-55.” Two adjacent sgRNA target sites were selected within the open reading frames of *SlTLP5* and *SlTLP6*, and PCR was used to introduce the target sequence downstream of the promoter and upstream of the sgRNA sequence. The sgRNA expression cassette was assembled into the gene-editing binary vector pYLCRISPR/Cas9P35S [23] using multiple rounds of overlapping PCR and introduced into the “S-55” plants using *Agrobacterium tumefaciens*-mediated transformation. PCR amplifications were performed using the DNA of positive tomato plants that were preliminarily selected through kanamycin screening. The mutation site and mutation type were analyzed by comparison with the original sequence. T0 mutant plants with early gene expression termination were selected for inbreeding. The pure strains obtained via screening and sequencing were infected with pathogens to identify the disease resistance of gene-edited tomato plants. The above primers can be found in Appendix A.

### 2.5. Disease Resistance of Transgenic Plants

Overexpressing and gene-edited transgenic plants were infected with the five pathogens, as described above (*R. solanacearum* and *C. michiganense* subsp. *michiganense* as noted previously) [24,25]. Infected plants were preserved in a growth chamber at 27 °C and wilting or symptoms were recorded on day 7. Wilting symptoms were scored using a grade of 0 to 4: (0) healthy plants without wilting; (1) plants with 25% withered leaves; (2) plants with 50% withered leaves; (3) plants with 75% withered leaves; and (4) plants with completely withered leaves. The disease index was calculated by averaging the disease scores for each plant in the experiment.

Fungal pathogens *F. oxysporum* [26,27], *V. dahliae*, and *F. solani* were quantified based on disease severity [28] and classified into five levels according to observations during pathogen invasion: (0) healthy plants (without obvious wilting or yellowing symptoms); (1) Cotyledon wilt or fall off; (2) plants with 30–50% true leaves withered or shed; (3) plants with 50–80% true leaves withered or shed; and (4) plants with all leaves shed or dead. The disease classification was scored 14 days after pathogen infection, using five individual plants for each treatment.

### 2.6. Determination of ROS Antioxidant Physiological Indexes in Transgenic Plants

The transgenic plants treated for 7 days were stained with diaminobenzidine (DAB)/nitroblue tetrazolium (NBT) to observe H_2_O_2_ (hydrogen peroxide) and O_2_^–^ accumulation in the leaves of the whole infected plant. The plants transfected with the pBI121 empty vector were used as a control for overexpressing plants, and off-target effector plants were used as a control for gene-edited plants [29]. Peroxidase (POD), superoxide dismutase (SOD), catalase (CAT), and ascorbate peroxidase (APX) activities were determined as reported previously [30].

### 2.7. Determination of β-1,3-GA Activity

In infected plants, β-1,3-GA can catalyze the hydrolysis of β-1,3-glucosidic bond, thus destroying the fungal cell wall. Therefore, the determination of β-1,3-GA activity is widely used in plant pathological studies. The activity of β-1,3-GA was determined as described by Zong et al. [31], and crude enzyme solution (0.03 mL) was pipetted into 0.07 mL of kelp toxin and incubated for 40 min at 37 °C. Then, 1.5 mL of 3,5-dinitrosalicylic acid reagent was added to a 100 °C water bath for 5 min, and the absorbance was measured at 540 nm. The glucose (mg) produced per hour by the breakdown of kelp toxin through the enzymatic reaction system was used as the enzyme activity unit.

### 2.8. Data Statistics

Each experiment was performed at least three times. The IBM SPSS Statistics 24 statistical software (IBM, Chicago, IL, USA) was used for data processing, and one-way analysis of variance was adopted for data analysis. Duncan’s test was used for post hoc analysis, and the difference at the 95% level was considered significant. Normality, homogeneity of variance, and data independence were assessed before all analyses, and the results are expressed as the mean ± standard deviation.

## 3. Results

### 3.1. Response of the SlTLP5 and SlTLP6 Genes in Five Soil-Borne Diseases

After infection with five pathogens, the expression patterns of *SlTLP5* and *SlTLP6* were detected using RT-qPCR on the “No. 5” and “S-55” inbred lines on day 0 (control) and days 3, 5, 7, and 9. In the “No. 5” susceptible cultivar, the transcript levels of *SlTLP5* and *SlTLP6* increased in those plants treated with three fungal pathogens and were very similar in those treated with the same pathogen. As for the bacterial pathogen treatment, plants treated with *R. solanacearum* exhibited a stronger response than those treated with *C. michiganense* subsp. *michiganense*, and *SlTLP5* and *SlTLP6* expression reached significant levels on day 3 after the *R. solanacearum* treatment, with very similar *SlTLP5* and *SlTLP6* transcript levels. The *SlTLP6* increased but did not reach significant levels after the *C. michiganense* subsp. *michiganense* treatment, and *SlTLP5* showed a decreasing trend (Figure 1A–E).

In the “S-55” disease-resistant cultivar, the transcript levels of the *SlTLP5* and *SlTLP6* genes increased after treatment with the three fungal pathogens, and the transcript levels were very similar in plants treated with the same pathogen. The transcript levels of the *SlTLP5* and *SlTLP6* genes both increased and reached significant levels with the bacterial pathogen treatment (Figure 2A–E).

After infection with the five pathogens, the transcription levels of the *SlTLP5* and *SlTLP6* genes in the disease-resistant cultivars were higher than those in the susceptible cultivars, and the transcription level of *SlTLP6* was slightly greater than that of *SlTLP5*.

### 3.2. Overexpression of TLP Conferred Increased Disease Resistance to Tomato Plants

Recombinant binary vectors pBI121-*SlTLP5* and pBI121-*SlTLP6* were constructed and used to transfect transgenic plants and an empty plasmid frame control (Figure 3A). The positive plants overexpressing *SlTLP5* and *SlTLP6* were detected using RT-qPCR. To identify disease resistance in the transgenic plants, transgenic tomato seedlings were inoculated with the five pathogens, as described in the Materials and Methods. We found that both transgenic lines exhibited increased resistance to all five diseases compared with the control plants, and they reached a significant level (Figure 4A,B). The *SlTLP5-* and *SlTLP6*-overexpressing lines demonstrated a stronger resistance to fungal diseases than bacterial diseases. After treatment with the bacterial pathogens, the control group withered severely, with disease indices of 3.1 and 3.2 for *R. solanacearum* and *C. michiganense* subsp. *michiganense*, respectively. The plants overexpressing *SlTLP5* and *SlTLP6* were less injured than the control group (Appendix A), with disease indices of only 1.8 and 2.2, respectively, and the lines overexpressing *SlTLP5* were slightly more resistant than those overexpressing *SlTLP6*.

### 3.3. TLP Knockout Reduces Disease Resistance in Tomato Plants

We assumed that *SlTLP5* and *SlTLP6* are required in multiple disease resistance. CRISPR/Cas9 *SlTLP5* and *SlTLP6* loss-of-function alleles were obtained in the resistant “S-55” variety. Twenty-eight transgenic plants were identified in the *SlTLP5* knockout, carrying 1- and 2-nucleotide deletions (referred to as KO-*SlTLP5*-Line11 and KO-*SlTLP5*-Line12, respectively), and thirty-one transgenic plants were identified in the *SlTLP6* knockout, carrying 3- and 5-nucleotide deletions (referred to as KO-*SlTLP6*-Line23 and KO-*SlTLP6*-Line25, respectively) (Figure 3B). Under normal growth conditions, the phenotypes of all the gene-edited plants were indistinguishable from those of the control plants.

The KO-*SlTLP5*-Line, KO-*SlTLP6*-Line, and off-target control plants were infected with the five pathogens. We found that *SlTLP5*-knockout- and *SlTLP6*-knockout line plants withered severely compared with the control plants (Figure 4C,D). The data revealed that both the *TLP5* and *TLP6* knockout plants had significantly higher disease severity than the control plants among those treated with the three fungal pathogens (Appendix A). Both the *SlTLP5* and *SlTLP6* knockout plants exhibited increased disease severity than the control plants among those treated with the bacterial pathogens, but significant levels were not reached, indicating that *SlTLP5* and *SlTLP6* are more sensitive to fungal pathogens.

### 3.4. Physiological Changes in Resistance of Overexpressing Plants of Susceptible Cultivars

After treatment with the five pathogens for 7 days, DAB and NBT staining revealed that there were fewer spots on the leaves of plants overexpressing *SlTLP5* and *SlTLP6* and more spots on the control plants (Figure 5A).

The enzyme activity of the antioxidant system was determined through its ROS levels. The transgenic lines overexpressing *SlTLP5* and *SlTLP6* showed the same trend under the fungal pathogen treatment. Compared with the control group, in transgenic lines overexpressing *SlTLP5* and *SlTLP6*, the activities of POD, SOD, APX, and CAT antioxidant enzymes did not increase but instead decreased, with the most significant decrease in CAT. However, the activities of APX, SOD, and POD decreased under the bacterial pathogen treatment (Appendix A), whereas CAT activity in response to bacterial pathogens was the opposite compared with that of the fungal pathogen treatment, and it increased significantly after CAT overexpression (Figure 6).

β-1,3-GA activity was determined in transgenic plants overexpressing *SlTLP5* and *SlTLP6* after the pathogen treatment. The β-1,3-GA activity in all the transgenic lines overexpressing *SlTLP5* and *SlTLP6* treated with the five pathogens significantly increased (Figure 7 and Figure 8) and was significantly higher than that of the control group. Notably, the transgenic lines overexpressing *SlTLP5* and *SlTLP6* had increased resistance to diseases, and this was probably due to increased β-1,3-GA activity.

### 3.5. Physiological Changes in Resistance of Knockout Plants of Disease-Resistant Cultivars

Seven days after infection, DAB and NBT staining revealed that there were more spots on the leaves of the positive *SlTLP5* and *SlTLP6* knockout plants, indicating that H_2_O_2_ and O_2_
^−^ accumulation caused more damage to them (Figure 5B). The enzyme activity of the antioxidant system was determined using ROS. We found that the changes in ROS enzyme activity in the *SlTLP5* and *SlTLP6* knockout lines were similar. In response to the three fungal pathogens, we found that there were no significant changes in POD, SOD, APX, or CAT activity after the knockout of *SlTLP5* and *SlTLP6*. In response to the two bacterial pathogens, there was no significant change in POD or SOD (Appendix A), and the activities of APX and CAT increased and decreased, respectively, after the knockout of *SlTLP5* and *SlTLP6*, the results of which were similar to those after fungal infection (Figure 7).

In the *SlTLP5* and *SlTLP6* lines, the activity of β-1,3-Ga decreased after treatment with the five pathogens; in particular, β-1,3-Ga activity in the *TLP5* lines was significantly lower than in the control and *TLP6* lines (Figure 8), indicating that the activity of β-1,3-Ga was affected by the knockout of the *TLP5* and *TLP6* lines, and thus, the plant resistance was reduced.

## 4. Discussion

It is important to understand the defense mechanism of plants to induce basic resistance under different stress conditions. *TLP*, a member of the PR-5 family, includes *TLP*, Osmotin, and Zeamatin [1]. The PR-5 family is associated with responses to biotic stress. *TLPs* [32] have been identified in more than 180 plants, including dicots, monocots, gymnosperms, bryophytes, and algae, and some *TLPs* demonstrate broad-spectrum resistance to a variety of pathogens. Their overexpression can enhance resistance to pathogens, including *Alternaria solani*, *Sclerotinia sclerotiorum*, *P. syringae* pv. DC3000, *Puccinia triticina*, and *Colletotrichum gloeosporioides* f. sp. *manitiss* [7,8,33,34]. In addition, *TLPs* can also be induced by bacterial pathogens, abiotic stresses (e.g., wounding, drought, osmotic stress, low temperature, high salt, and UV radiation), and phytohormones [35,36,37]. In this study, the susceptible tomato cultivar “No. 5” and the resistant tomato cultivar “S-55” were infected with five soil-borne diseases. According to the RT-qPCR results, *SlTLP5* and *SlTLP6* were expressed to different degrees in “No. 5”, and the cultivars that were treated with the three fungi had higher *SlTLP5* and *SlTLP6* levels than those infected with *C. michiganense* subsp. *michiganense* and *R. solanacearum* (Figure 1). *SlTLP5* and *SlTLP6* were significantly increased in the disease-resistant tomato cultivar “S-55”. Similarly, in the disease-resistant plants, those treated with three different pathogenic fungi had higher *SlTLP5* and *SlTLP6* levels than those infected with *C. michiganense* subsp. *michiganense*, or *R. solanacearum*. *TLP* was more sensitive to fungal diseases, and *SlTLP6* transcript levels were higher than those of *SlTLP5* (Figure 2).

The anti-fungal effect of *TLP* has been applied to the breeding of a variety of crops. For example, *CkTLP* significantly enhanced resistance to *V. dahliae* in Arabidopsis [38]. Cold-induced *taTLP* accumulated in apoplasts contributes to the resistance of winter wheat to *Microdochium nivale* [39]. Additionally, the antibacterial activity of *TLP* was also found in rice [40] and potatoes [41]. Overexpression lines and knockout lines were generated in “No. 5” and “S-55” (Figure 7). The overexpression lines had enhanced disease resistance after treatment with the five pathogens, and the knockout lines had reduced disease resistance (Figure 8) and stronger resistance to fungal diseases than to bacteria.

Fungal-mediated biotic stress activates the plant immune system by sensing pathogen-associated molecular patterns and molecular receptors [42]. Subsequently, ROS are formed to induce abscisic acid, salicylic acid, and jasmonic acid signals and upregulate PR genes [43]. Tobacco *TLPs* determine the morphology of cell death through a critical role in the RAS2/cAMP-mediated regulation of intracellular ROS accumulation [44]. It was hypothesized that ROS in tomato plants are involved in the disease resistance mechanism of *SlTLP5* and *SlTLP6*, and the data revealed that the overexpression of *SlTLP5* and *SlTLP6* in the “No. 5” cultivar treated with the fungal pathogens decreased APX, CAT, POD, and SOD activities toward ROS. There was a slight difference between those treated with bacterial pathogens and those treated with fungi, in that CAT increased in those treated with the bacterial pathogens (Figure 6). *SlTLP5* and *SlTLP6* knockout experiments in the resistant “S-55” cultivar revealed that there was little difference in APX, POD, and SOD activities toward ROS (Figure 7), and CAT activity increased and decreased in fungal and bacterial diseases, respectively, indicating that ROS possessed different regulatory pathways in the *TLP* disease resistance mechanism. Generally, APX, POD, and SOD activities were negatively correlated with *SlTLP5* and *SlTLP6* in the five soil-borne diseases, and CAT was positively and negatively correlated in the bacterial and fungal diseases, respectively.

Plant *TLPs* inhibit pathogenic and non-pathogenic fungi mainly by lysing fungal spores and inhibiting spore germination and growth patterns [45,46]. This probably occurs because of intra-β-1,3-GA activity in the acidic domain space of *TLPs* [47] and their role in inducing mechanisms involved in pathogen defense, including generating phenylpropanoid and phytoalexin [48,49]. Chitinase, β-1,3-GA, and miRNAs are involved in the positive regulation (miR164a, miR168a, and miR393) and negative regulation (miR394) of Fusarium resistance in Allium plants [50,51,52,53]. After *Elsinoe ampelina* inoculation, the TLP gene exhibited increased expression, as did chitinase and antibacterial protein genes such as β-1,3GA [54,55], PR1/PR1a [56], polygalacturonase inhibitor protein [57], and lipid transfer protein [58]. Likewise, β-1,3-GA activity was determined and we found that it significantly increased in both *SlTLP5*- and *SlTLP6*-overexpressing transgenic plants treated with all five pathogens, regardless of pathogen species, and it was stronger in *SlTLP5*-overexpressing plants than in plants overexpressing *SlTLP6* (Figure 8). β-1,3-GA activity decreased in the *SlTLP5* and *SlTLP6* knockout transgenic lines and decreased significantly in the *SlTLP5* knockout plants treated with the five pathogens, suggesting that the expression of *TLP* is associated with β-1,3-GA activity and that it positively regulated β-1,3-GA activity to confer resistance to the five tomato pathogens.

Although the *TLP* family has been extensively studied, the core mechanism of resistance, particularly upstream regulation, remains unclear. However, the mechanism of *SlTLP5* and *SlTLP6* disease resistance can provide a reference for subsequent studies. In other words, the ROS pathway feedback was different under fungal and bacterial infections. The expression of *TLP* inhibited POD, SOD, APX, and CAT activities under fungal infection. POD, SOD, and APX activities were similar to those during bacterial infection, whereas CAT activity increased. Although the RT-qPCR results indicated that *SITLP6* transcript levels were higher than those of *SITLP5*, *SITLP5* provided slightly more resistance than *SITLP6* according to the symptom score, which is consistent with previous studies [18]. This may be due to higher β-1,3-GA activity in plants overexpressing *SITLP5* than in those overexpressing *SITLP6*. In conclusion, *TLP* confers plant resistance through β-1,3-GA activity under pathogen infection. The resistance conferred by *SITLP5* and *SITLP6* to the five pathogens makes them ideal candidates for plant transformations aimed at producing resistant crops.

## Figures and Tables

**Figure 1 genes-14-01622-f001:**
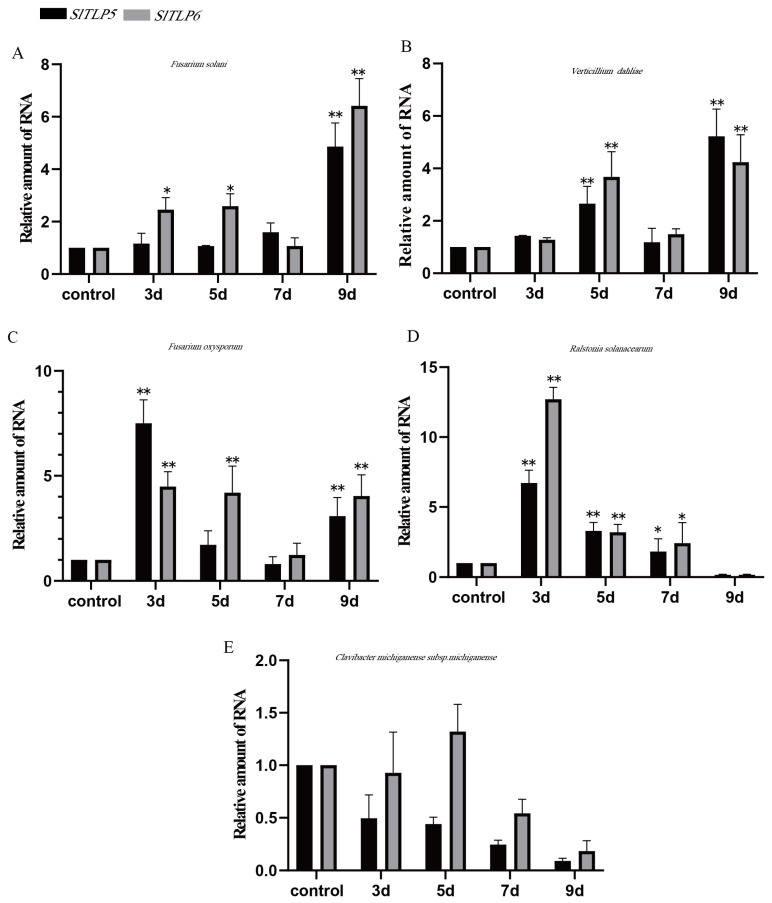
Response of *SlTLP5* and *SlTLP6* genes in susceptible cultivar “No. 5”. Total RNA was extracted from seedlings at four infection stages, days 0, 3, 5, 7, and 9, and expression of the genes was assessed using RT-qPCR. (**A**–**E**) represent *F. solani*, *V. dahliae*, *F. oxysporum*, *R. solanacearum*, and *C. michiganense* subsp. michiganense treatments, respectively. Note: The asterisk indicates a significant difference compared to the control group (* *p* < 0.05 and ** *p* < 0.01). Error bars indicate standard deviations (SDs) for three replicates.

**Figure 2 genes-14-01622-f002:**
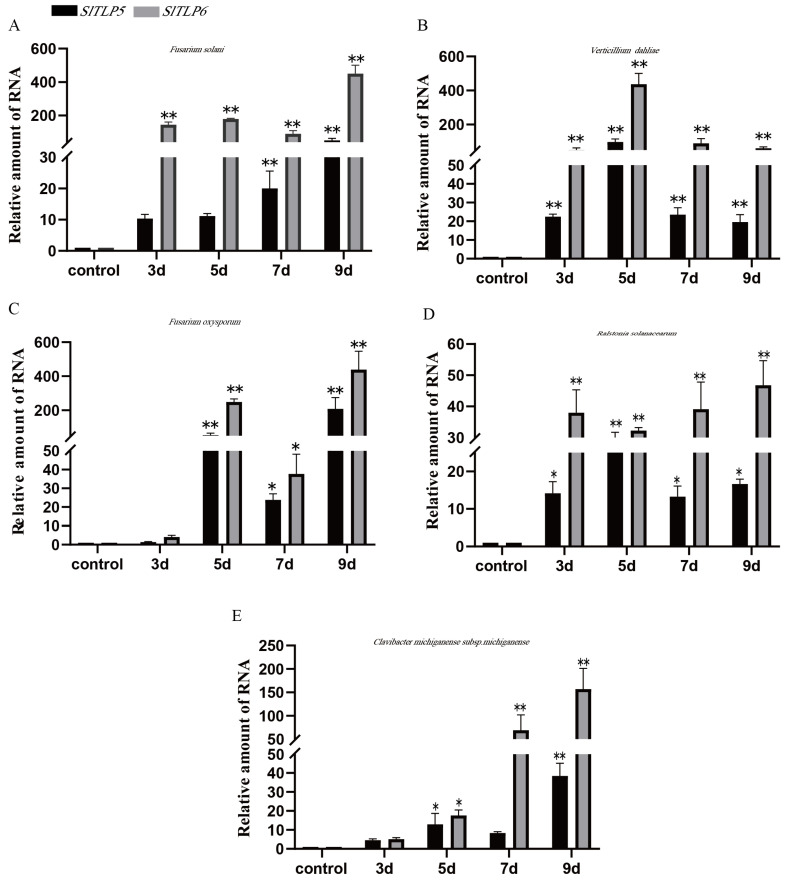
Response of *SlTLP5* and *SlTLP6* in the resistant cultivar “S-55”. Total RNA was extracted from seedlings at four infection stages, days 0, 3, 5, 7, and 9, and expression of genes was assessed using RT-qPCR. (**A**–**E**) represent *F. solani*, *V. dahliae*, *F. oxysporum*, *R. solanacearum*, and *C. michiganense* subsp. *michiganense* treatments, respectively. Note: The asterisk indicates a significant difference compared to the control group (* *p* < 0.05 and ** *p* < 0.01). Error bars indicate SDs for three replicates.

**Figure 3 genes-14-01622-f003:**
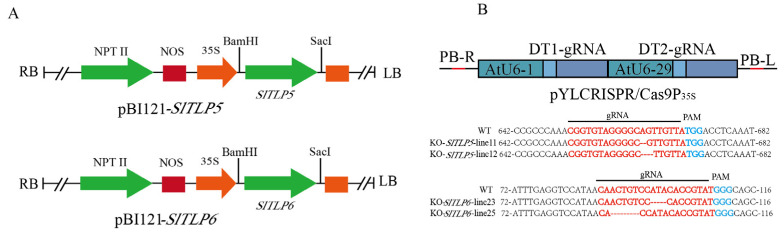
Generation of transgenic plants. (**A**) *SlTLP5* and *SlTLP6* overexpression vector backbone construction. (**B**) Schematic representation of CRISPR/Cas9 cassettes used for *SlTLP5* and *SlTLP6* mutant vectors. The plant genome sequences of *SlTLP5* and *SlTLP6* were aligned via CLUSTALX nucleic acid sequence alignment, and gRNA sequences are highlighted in red.

**Figure 4 genes-14-01622-f004:**
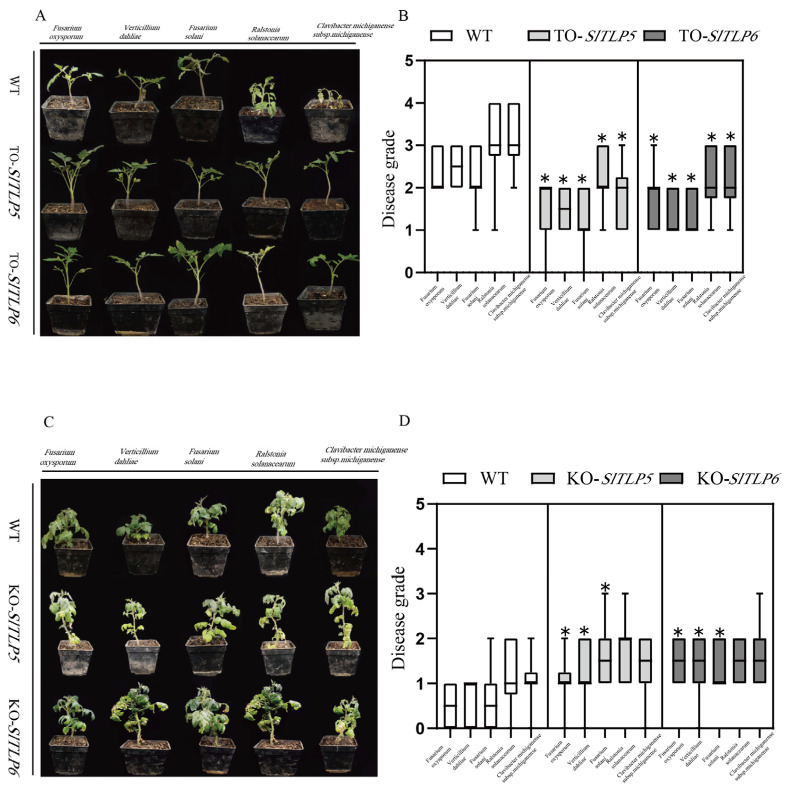
Gene function validation via overexpression and CRISPR/Cas9 gene editing. Plants treated with bacterial and fungal pathogens were photographed on days 7 and 14 after infection for wilting disease and disease resistance identification. (**A**) Plants overexpressing *SlTLP5* and *SlTLP6* demonstrated increased disease resistance. (**B**) Disease grades (* indicates *p* < 0.05) for To-*TLP5*, To-*TLP6*, and empty plasmid transgenic plants after infection with five pathogens. (**C**) *SlTLP5* and *SlTLP6* knockouts decreased plant disease resistance. (**D**) Disease grading for KO-*SlTLP5*, KO-*SlTLP6*, and off-target transgenic plants after infection with the five pathogens. The error bar is the standard deviation of three biological replicates: Student’s *t*-test; asterisks (*) indicate *p* < 0.05.

**Figure 5 genes-14-01622-f005:**
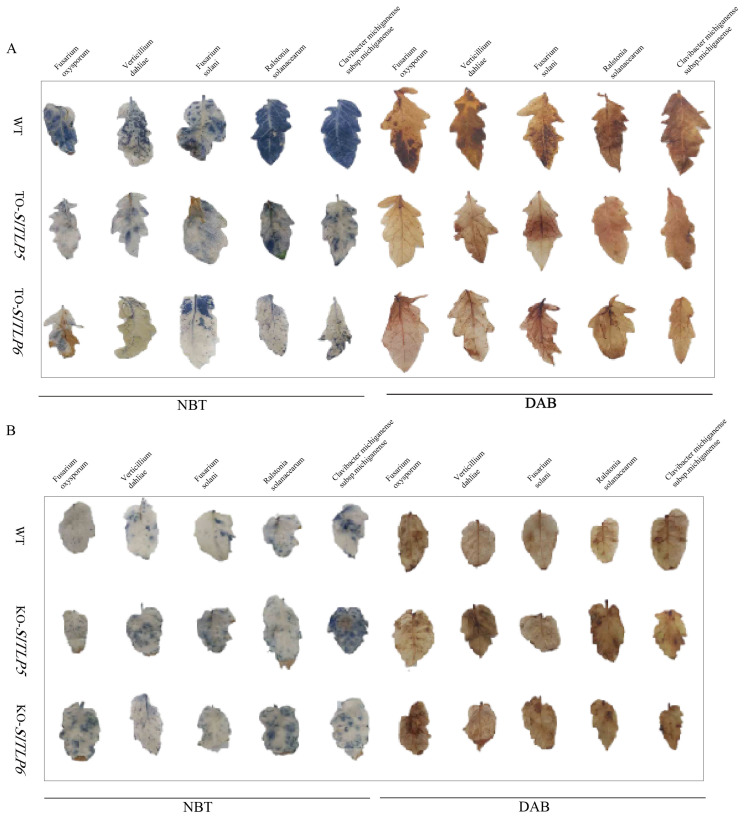
H_2_O_2_ and O_2_^−^ accumulation in leaves. (**A**) H_2_O_2_ and O_2_^−^ accumulation in leaves reduces in overexpressing transgenic plants. (**B**) H_2_O_2_ and O_2_^−^ accumulation increases in the leaves of knockout plants.

**Figure 6 genes-14-01622-f006:**
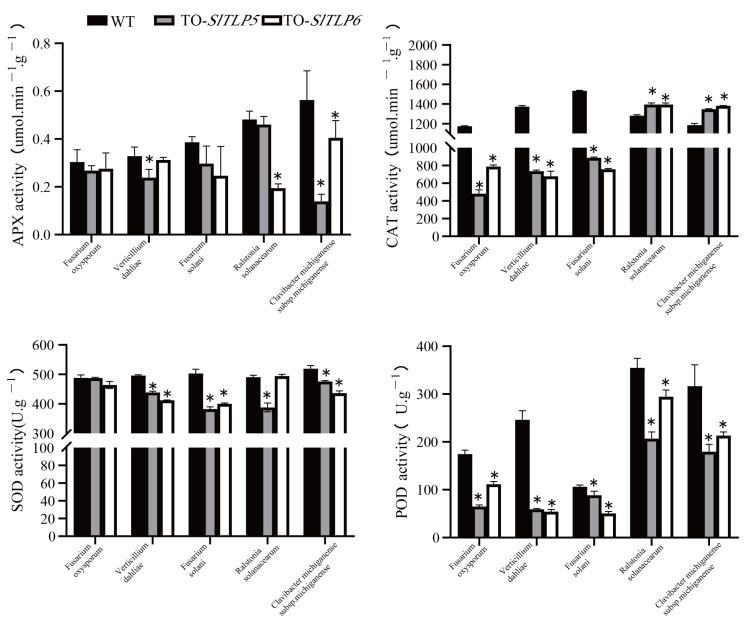
Changes in physiological parameters of overexpressing transgenic plants infected with pathogens. The error bar is the standard deviation of three biological replicates: Student’s *t*-test; asterisks (*) indicate *p* < 0.05.

**Figure 7 genes-14-01622-f007:**
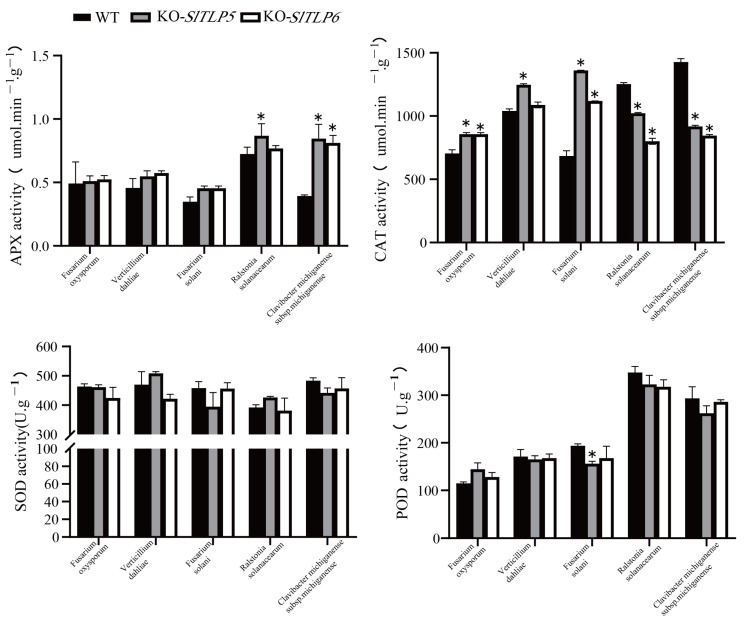
Changes in physiological parameters of knockout transgenic plants infected with pathogens. The error bar is the standard deviation of three biological replicates: Student’s *t*-test; asterisks (*) indicate *p* < 0.05.

**Figure 8 genes-14-01622-f008:**
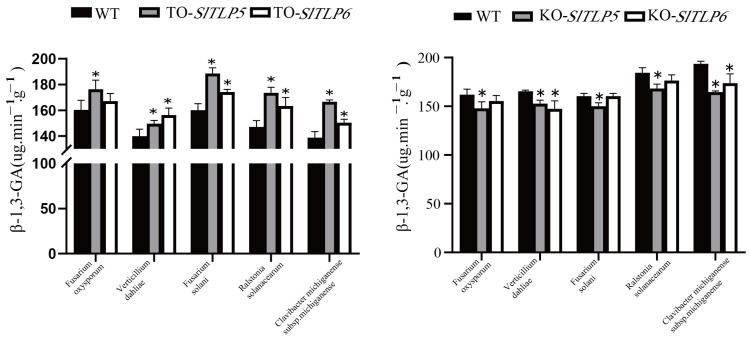
Changes in β-1,3-glucanase activity in overexpressing and knockout transgenic plants. The error bar is the standard deviation of three biological replicates: Student’s *t*-test; asterisks (*) indicate *p* < 0.05.

## Data Availability

Not applicable.

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
