# Peer review of "Tomato-Thaumatin-like Protein Genes Solyc08g080660 and Solyc08g080670 Confer Resistance to Five Soil-Borne Diseases by Enhancing β-1,3-Glucanase Activity"

_genes, 2023, doi:10.3390/genes14081622_

Round 1
Reviewer 1 Report
Please read all the comments, I get long winded.
The Abstract is too detailed
The Introduction or 1st paragraph lacks the essential information about the overall experimental design
The Results are - poorly written and confusing - and so difficult to follow.
The Discussion is well written and clear.

I think there are problems with the English. Follow the comments and changes and please modify the rest of the text to clarify what was actually done.
Author Response
Dear Reviewer,
Thank you for your decision and constructive comments on my manuscript. We have carefully considered the reviewers' suggestions and made some changes. We tried our best to improve and made some revisions to the manuscript.
Yellow section modified according to your comments. Point-to-point revision instructions are as follows:
Point 1: The abstract is too detailed.
Response 1: In the Abstract section, we reorganized and cut (line 36) without changing the meaning of the original text.
Point 2: The Introduction or paragraph 1 lacks basic information about the overall experimental design.
Response 2: In the last paragraph of the Introduction, we introduced the experimental information and removed redundant results (Line 84).
Point 3: The result is - poorly written, confusing - hard to understand.
Response 3: We apologize for the error and reorganized the semantic logic of the article (lines 205 and 334). Additionally, we have embellished the article at https://www.mdpi.com/authors/english to make it easier for reviewers and editors to read.
We would like to thank the referees again for taking the time to review our manuscript.
Reviewer 2 Report
General information:
This is an exciting manuscript investigating the function of two TLPs for tomato resistance against bacterial and fungal pathogens. The study is well-planned and correctly presented. There are however some issues regarding the preparations of this manuscript that in my opinion must be corrected before the article can be published.
The introduction should not contain results but be a short description of the current state of knowledge on the subject. The figures are often of low resolution and higher quality graphics should be included. The writing of scientific names or gene names should be checked and corrected. The results used to prepare graphics should be included in the supplementary file and the results confirming RNA purity and no phenotypic differences of the obtained tomato lines. Additionally, the scoring system does not produce parametric data, and such data should be treated as non-parametric, please do not compare mean values and do not use parametric statistical tests.
In text comments
Line 10: please add the descriptor abbreviation to the scientific names of plants in the case of tomato Mill. L. and write scientific names in italics
Line 99 remove or seven
Line 110 (w/v) agar
Line 154 for non-parametric scoring system median or modal value should be used instead of mean
Line 180 Please do not use statistics for parametric data on non-parametric values (scoring system)
Figure 1 is quite blurry please replace it with higher-quality graphics
For figures 1 and 2 please add the pathogen names above the plots
Figure 2 Please use italics for scientific names
Line 249 Please add the pictures of obtained lines to the supplementary file
Figure 4 please provide a higher-quality version of this figure and write scientific names in italics
Line 339 and others please provide at least one citation
Please add the required statements at the end of the manuscript
Author Response
Dear reviewer,
Thank you for your decision and constructive comments on my manuscript. We have carefully considered the suggestion of Reviewer and make some changes.
We think carefully about your comments,Revisionnotes, point-to-point, are given as follows:
Point 1:Line 10: please add the descriptor abbreviation to the scientific names of plants in the case of tomato Mill. L. and write scientific names in italics
Response 1: Descriptor abbreviations have been added and written in italics (line 10).
Point 2:Line 99 remove or seven
Response 2: We apologize for the error and corrected it(line 112).
Point 3:Line 110 (w/v) agar
Response 3: We apologize for the error and corrected it(line 124).
Point 4:Line 154 for non-parametric scoring system median or modal value should be used instead of mean
Response 4: I am sorry that this part was not clear in the original manuscript. First, our fungal and bacterial pathogen symptoms were very similar to those in previous studies, so we referred to their scoring system. Second, we chose different scoring criteria based on differences in fungal and bacterial symptoms. For bacterial diseases, we refer to [25][26] and for fungal diseases, we refer to [27-29]. The disease index was calculated by averaging the disease scores of each plant in the experiment. The calculation method was shown in the literature (line 164).
Point 5:Line 180 Please do not use statistics for parametric data on non-parametric values (scoring system)
Response 5:Thank you so much for your careful check. Tomato severity during pathogen invasion experience was divided into different grades. After inoculation, the disease was scored and calculated according to the calculation method described in the previous published paper(line 197).
Point 6:Figure 1 is quite blurry please replace it with higher-quality graphics
Response 6: We have replaced it with a higher-quality image(line 227).
Point 7:For figures 1 and 2 please add the pathogen names above the plots
Response 7: We add pathogen names to Figures 1 and 2(line 227 and line 235).
Point 8:Figure 2 Please use italics for scientific names
Response 8: We apologize for the error and corrected it(line 235).
Point 9:Line 249 Please add the pictures of obtained lines to the supplementary file
Response 9: We have added the strain picture to the supplementary file.
Point 10:Figure 4 please provide a higher-quality version of this figure and write scientific names in italics
Response 10: We'll replace it with a higher quality image and write scientific names in italics(line 281).
Point 11:Line 339 and others please provide at least one citation
Response 11: We removed "and so on" and added a reference to "potatoes"(line 374).
In addition, in order to make the comment experts and editors are easier to read, we touch up articles at https://www.mdpi.com/authors/english.
We would like to thank the referee again for taking the time to review our manuscript.
Round 2
Reviewer 1 Report
Looks like I overlooked this statement somehow "significant differences between the plants and corresponding control plants under the
same treatment ".
The English reads much better now and I see that you found lots of typos all over the place too. I say after minor revisions, just to make sure to go over it one more time, since whoever was hurrying left Word comments in the uploaded file. Might find other small errors.
I google translated the response! Worked.
****The publication relies on the presentation of the data. The figure resolution is very very low that even when zooming it was nearly impossible to read anything. This is the attached file with a 430% zoom, still cant read anything properly. I highly suggest that this is remedied.

Author Response
Dear reviewer,
Thank you again for your decision and constructive comments on my manuscript. We have carefully considered the Reviewer's suggestion, and we have done our best to revise the manuscript to improve the quality of the paper.
Point 1:Looks like I overlooked this statement somehow "significant differences between the plants and corresponding control plants under the same treatment ".
Response 1: Thank you for reminding us that this sentence does read confusing, we have combed and simplified this sentence and changed it to "The asterisk indicates a significant difference compared to the control group".
Point 2:The English reads much better now and I see that you found lots of typos all over the place too. I say after minor revisions, just to make sure to go over it one more time, since whoever was hurrying left Word comments in the uploaded file. Might find other small errors.
Response 2: We double-checked the article for typos and other possible errors.
Point 3:The publication relies on the presentation of the data. The figure resolution is very very low that even when zooming it was nearly impossible to read anything. This is the attached file with a 430% zoom, still cant read anything properly. I highly suggest that this is remedied.
Response 3: We have replaced the image you mentioned with a high-definition image, and we have also replaced other blurred images.
We would like to thank the referee again for taking the time to review our manuscript.
Reviewer 2 Report
I would like to thank the authors for properly responding to most of my comments. However, I still feel that the authors should include more supplementary data for the study including raw data used for graphs preparation and data on RT-qPCR. Additionally, the introduction to the manuscript in my opinion should contain more background for undertaken study and should not present new data. Since the high quality of the presented study, I can consider this manuscript acceptable for publication in Genes in its current state.
Author Response
Dear reviewer,
Thank you again for your decision and constructive comments on my manuscript. This time, we added experimental data, including RT-qPCR, disease resistance and enzyme activity determination of transgenic plants. Secondly, we have sorted out the introduction and deleted the redundant data. Finally, we also check the paper for possible typos and other errors in order to improve the quality of the paper.
We would like to thank the referee again for taking the time to review our manuscript.